

# Enhanced transformer for length-controlled abstractive summarization based on summary output area

Yusuf Sunusi, Nazlia Omar and Lailatul Qadri Zakaria

Center for Artificial Intelligence Technology, Faculty of Information Science and Technology, Universiti Kebangsaan Malaysia, Bangi, Selangor, Malaysia

## ABSTRACT

Recent advancements in abstractive summarization models, particularly those built on encoder-decoder architectures, typically produce a single summary for each source text. Controlling the length of summaries is crucial for practical applications, such as crafting cover summaries for newspapers or magazines with varying slot sizes. Current research in length-controllable abstractive summarization employs techniques like length embeddings in the decoder module or a word-level extractive module in the encoder-decoder model. However, these approaches, while effective in determining when to halt decoding, fall short in selecting relevant information to include within the specified length constraint. This article diverges from prior models reliant on predefined lengths. Instead, it introduces a novel approach to length-controllable abstractive summarization by integrating an image processing phase. This phase determines the specific size of the summary output slot. The proposed model harnesses enhanced T5 and GPT models, seamlessly adapting summaries to designated slots. The computed area of a given slot is employed in both models to generate abstractive summaries tailored to fit the output slot perfectly. Experimental evaluations on the CNN/Daily Mail dataset demonstrate the model's success in performing length-controlled summarization, yielding superior results.

## INTRODUCTION

In recent years, the demand for utilizing data from diverse sources, including scientific literature, medical reports, and social networks, has surged. Text summarization, a crucial process, involves generating a concise, coherent summary of lengthy text documents. Automatic text summarization employs two primary approaches, extractive and abstractive, depending on how the summary is generated. Extractive approaches (*Duc et al., 2016*; *Krishnan et al., 2019*; *Palliyali et al., 2021*; *Saziyabegum & Sajja, 2016*) directly extract explicit words and phrases from the source text. In contrast, abstractive approaches (*Badgujar, Jethani & Ghorpade, 2018*; *Fan, Grangier & Auli, 2018*; *Palaskar et al., 2019*; *Wan et al., 2018*) generate novel words and phrases not present in the source text, akin to human-written summaries.

Abstractive text summarization aims to produce a shorter version of the input text while preserving its meaning (*Abdelwahab, Al Moaiad & Abu Bakar, 2023*; *Wang & Ren, 2021*).

Corresponding author
Yusuf Sunusi,
p103490@siswa.ukm.edu.my

This method involves generating abstracts through human-like thinking, requiring the model to possess high capabilities in characterizing, understanding, and generating (*Fejer & Omar, 2014*; *Abolohom & Omar, 2015*; *Alshalabi, Omar & Tiun, 2017*). Commonly employed in abstractive summarization methods, the Seq2Seq model based on recurrent neural networks (including LSTM, GRU, *etc.*) (*Hao et al., 2020*; *Kouris, Alexandridis & Stafylopatis, 2021*; *Wang et al., 2018*; *Zhang et al., 2019*) follows an encoder-decoder structure. The encoder encodes the target text into a context vector as a new representation, while the decoder extracts information from the context vector and decodes it into an abstractive summary (*Schumann, 2018*). While this approach produces coherent summaries closely matching the source text, it faces challenges in processing long texts due to the limitations of long-term dependence. For document-level, long sequence summarization tasks especially, it may lose key information and generate redundant content. The encoder-decoder structure with an attention mechanism (*Tang et al., 2018*; *Yang, Tang & Tang, 2018*; *Yolchuyeva, Németh & Gyires-Toth, 2020*), successfully applied to abstractive text summarization tasks (*Nguyen, Le & Tran, 2020*; *Omar & Al-Tashi, 2018*), has addressed some of these challenges. *Blekanov, Tarasov & Bodrunova (2022)* explored transformer-based neural network models, including LongFormer and T5, comparing them against BART on real-world data from Reddit. Additionally, synthetic data usage, a common approach in the machine translation domain under low-resource conditions, was adopted to enhance translation quality (*Hoang et al., 2018*). The iterative back-translation strategy, involving multiple rounds of training back-translation systems, has also been proven effective in machine translation (*Hoang et al., 2018*).

Transfer learning has revolutionized the field of natural language processing (NLP) by enabling models to leverage knowledge from one domain and apply it to another, often with limited data. This approach has proven particularly effective in tasks such as text summarization, where pre-trained models are fine-tuned on summarization-specific datasets to achieve state-of-the-art performance. For instance, *Raffel et al. (2020)* introduced a unified text-to-text framework that demonstrated the versatility of transfer learning across a range of language tasks, including summarization. Similarly, recent studies have shown that transfer learning can significantly improve the quality of abstractive text summarization, producing summaries that are both fluent and comprehensive (*Alomari et al., 2023*). By pre-training on extensive language models and then adapting these models to the summarization task, researchers have been able to capture intricate patterns in data without relying on handcrafted features, leading to more human-like and concise summaries (*Zhang, 2023*). The success of transfer learning in text summarization underscores its potential to enhance the efficiency and effectiveness of automated summarization systems, making them more accessible and useful in practical applications.

In the realm of abstractive summarization, the oversight of constraining summary length is now recognized as a critical factor. This is especially evident when considering diverse display scenarios, such as mobile devices or fixed advertisement slots on websites, where editors may require shorter summaries for optimal presentation. Unfortunately, existing abstractive summarization models lack training to accommodate such length

constraints (*Liu, Luo & Zhu, 2018*). *Fan, Grangier & Auli (2018)* addressed multi-sentence summarization using a convolutional sequence-to-sequence model, employing predefined and fixed special markers to represent length ranges. However, this method is limited to generating summaries within predefined length ranges, falling short of meeting arbitrary length constraints precisely. *Li et al. (2017)* extended the seq2seq framework with a generative model but neglected recurrent dependencies, limiting its representation capacity. Editorial needs for fitting summaries into specific cover slots further emphasize the importance of controllable summarization. Although length embeddings guide where to conclude decoding, they lack the ability to decide on relevant information within length constraints (*Saito et al., 2020*). Importantly, existing models do not address output area-based summarization which is crucial for scenarios like summarizing lengthy newspaper stories to fit a specific portion of a newspaper cover.

Our work aims to fill this gap by developing an arbitrary length-controllable abstractive text summarization model, enabling automatic summarization based on output area constraints while preserving summary quality. The primary contributions of this article are as follows:

- Introduction of output area-based summarization: We propose a novel approach for dynamically determining summary length based on the output medium area, ensuring that summaries are contextually appropriate and fit within designated spatial constraints.
- Development of length-controlled summarization models: We fine-tuned state-of-the-art versions of T5 (OSum) and GPT (OSumGPT) models to generate summaries using dynamically determined length constraints, offering flexibility and adaptability in automatic summarization tasks.

## RELATED WORK

*Kikuchi et al. (2016)* pioneered the use of length embeddings for length-controlled abstractive summarization, introducing both decoding-based and learning-based methods. *Fan, Grangier & Auli (2018)* similarly employed length embeddings at the start of the decoder module for length control, presenting a neural summarization model with a straightforward yet powerful mechanism for users to specify high-level attributes, shaping final summaries to better suit their requirements. *Liu, Luo & Zhu (2018)* proposed a CNN-based length-controllable summarization model, taking the desired length as input to the initial state of the decoder. *Takase & Okazaki (2019)* introduced positional encoding in a Transformer-based encoder-decoder model, representing the remaining length at each decoder step. *Saito et al. (2020)* adopted an extractive-and-abstractive summarization approach, integrating an extractive model into an abstractive encoder-decoder model.

As depicted in Table 1, various past works had achieved control over summary length, either with pre-defined constraints (*Fan, Grangier & Auli, 2018*; *Kikuchi et al., 2016*; *Zhang et al., 2019*) or arbitrary constraints (*Saito et al., 2020*; *Takase & Okazaki, 2019*; *Makino et al., 2019*). Despite advancements in enhancing length-constrained summarization quality, these models share the requirement for a specific length to be provided before

**Table 1 Summary of various text summarization techniques and their limitations.**

| Author | Technique | Length control | Limitations |
| --- | --- | --- | --- |
| *Fan, Grangier & Auli (2018)* | Convolutional Seq2Seq model | Yes | No emphasis on important words |
| *Zhang et al. (2019)* | CNN Seq2Seq model | Yes | Predefined summary length |
| *Lu et al. (2019)* | Long short-term memory (LSTM) | No | No summary length control |
| *Zhang et al. (2019)* | Convolutional Seq2Seq model | No | No summary length control |
| *Parida & Motlicek (2019)* | Transformer model | No | No summary length control |
| *Makino et al. (2019)* | CNN based encoder decoders | Yes | Loss of important words |
| *Takase & Okazaki (2019)* | Neural encoder-decoder model, Transformer | Yes | Pre-defined summary length |
| *Saito et al. (2020)* | Pointer-generator, Prototype extraction | Yes | Pre-defined summary length |
| *Song, Feng & Jing (2021)* | Reinforced abstractive summarization with ALCO | Yes | Balancing length and content preservation |
| *Fein & Cuevas (2022)* | Data augmentation for unsupervised summarization | Yes | Less effective than fine-tuning human summaries |
| *Dugar et al. (2022)* | Adversarial autoencoder model | Yes | Lower performance on non-English datasets |
| *Liu, Jia & Zhu (2022)* | Length-aware attention mechanism (LAAM) | Yes | Requires summary length balanced dataset |
| *Kumar et al. (2022)* | Extractive and abstractive summarization | Yes | Dependent on quality of extractive step |
| *Kwon, Kamigaito & Okumura (2023)* | Finetuning a pre-trained language model | Yes | Limited adaptability in diverse ATS contexts |

generating a summary. In *Saito et al. (2020)*, the specific length of the prototype text must be defined before inputting it into their encoder-decoder model for summary generation. Similarly, in *Takase & Okazaki (2019)*, the remaining length must be defined at each decoder step of the Transformer-based encoder-decoder model.

*Fein & Cuevas (2022)* introduced ExtraPhraseRank, a novel data augmentation strategy for unsupervised abstractive summarization that addresses the challenge of output length control. Their approach first employs TextRank to extract key sentences and then uses back-translation to enrich the diversity of expressions. Although the method showed improvements in ROUGE scores, it was found less effective than fine-tuning with human-written summaries, particularly in multi-document tasks such as summarizing collections of Amazon reviews.

*Dugar et al. (2022)* explored unsupervised abstractive text summarization with a focus on length control using an adversarial autoencoder model. This model encodes input into a smaller latent vector and decodes it to generate a concise summary, with the length controlled by the latent space dimensionality. The model was evaluated across various datasets, including Amazon and Yelp reviews, and demonstrated a ROUGE-1 score of around 24% for English datasets and 12% for Hindi, indicating its potential for multilingual summarization tasks.

*Liu, Jia & Zhu (2022)* proposed a length-aware attention mechanism (LAAM) that adapts the encoding of the source document based on the desired summary length. Unlike previous models that control length at the decoding stage, LAAM adjusts the encoding

process, enabling the generation of high-quality summaries even at lengths not seen in the training data. This method represents a significant advancement in length-controlled summarization by pre-training information selection (*Liu, Jia & Zhu, 2022*). The AINLPML team presented an end-to-end extractive and abstractive summarization approach for scientific documents in 2022. Their method, which fine-tuned a pre-trained BART model, outperformed baselines significantly, showcasing the effectiveness of combining extractive and abstractive methods for summarization tasks (*Kumar et al., 2022*).

*Song, Feng & Jing (2021)* introduced an Adaptive Length Controlling Optimization (ALCO) method that leverages reinforcement learning for abstractive summarization. ALCO integrates length constraints during sentence extraction and employs a saliency estimation mechanism to preserve essential information. This method has shown promise in balancing length constraint and content preservation, outperforming popular baselines on benchmark datasets.

A drawback of employing predefined or arbitrary length constraints in text summarization is the limitation when the summary is intended for a specific space or area. This is particularly relevant for applications like magazine or newspaper covers, where editors seek summaries tailored to fit within designated slots. The existing state-of-the-art approaches, including the work by *Kwon, Kamigaito & Okumura (2023)* and other preceding studies, do not specifically cater to the generation of summaries based on output area constraints where the requirement is to fit the summary into a predefined space, such as a designated slot on a magazine or newspaper cover. Despite the advancements made by various models in length-controlled summarization, a significant limitation remains in their ability to adapt to specific spatial constraints of the output medium. Most existing approaches, such as those by *Fein & Cuevas (2022)*, *Dugar et al. (2022)*, and *Liu, Jia & Zhu (2022)*, focus on controlling the summary length by either modifying the decoding process or adjusting the latent space dimensionality. However, these methods do not account for the actual physical space or area where the summary will be displayed, which is crucial for applications like magazine layouts, mobile interfaces, or dynamic web content. The absence of a mechanism to tailor summaries to fit within designated visual or spatial constraints means that these models may produce outputs that either overflow or leave unused space, detracting from the usability and aesthetic value of the final product. This limitation underscores the need for novel approaches that integrate length control with output area considerations, ensuring that generated summaries are not only accurate and relevant but also optimized for the specific medium in which they will be presented.

## MATERIALS AND METHODS

Our methodology comprises two primary components: area-based summary length prediction and fine-tuning pre-trained models for length-controlled summarization. In the first part, we developed a robust model for predicting the optimal summary length based on the available area of the output medium. This model utilizes computer vision techniques to analyze the medium's image, accurately identifying and measuring the

designated area for the summary. By calculating the available space, the model predicts an appropriate length for the summary, ensuring it fits within the specified spatial constraints. This approach enhances the coherence and relevance of the generated summaries, making them suitable for various contextual requirements, such as display constraints on different devices or specific editorial needs.

The second part of our method involves fine-tuning state-of-the-art pre-trained models, specifically T5 and GPT, to generate summaries that adhere to the predicted length. T5 and GPT models were selected for fine-tuning due to their proven effectiveness and versatility in various natural language processing tasks, including summarization. The T5 model, which stands for "Text-To-Text Transfer Transformer" is particularly well-suited for tasks that require converting one type of text input into another, such as generating summaries from articles. Its ability to handle diverse text-based tasks in a unified framework makes it an ideal choice for abstractive summarization. The GPT model, on the other hand, is renowned for its generative capabilities, particularly in producing coherent and contextually rich text. GPT-4, in particular, is among the most advanced language models, capable of understanding and generating human-like text across a wide range of topics. The decision to use these models was driven by their state-of-the-art performance in language generation and their ability to be fine-tuned effectively on specific tasks, such as those within the scope of this study. Additionally, their widespread adoption in the research community ensures robust support and continued improvement, making them reliable choices for achieving high-quality summarization outcomes.

We adapt these models to incorporate the predicted length as a guiding parameter during the summarization process. This involves modifying the input representation and decoding mechanisms to enforce length constraints dynamically while maintaining the quality and informativeness of the summaries. The fine-tuning process leverages transfer learning, allowing the models to adjust to the summarization task effectively by utilizing knowledge acquired from extensive pre-training on diverse datasets. By combining accurate length prediction with advanced generative capabilities of T5 and GPT, our approach aims to produce concise and contextually appropriate summaries.

Unlike traditional summarization approaches that primarily focus on controlling the length of summaries through *post-hoc* adjustments or predefined constraints, our method uniquely integrates spatial considerations from the outset. By combining area-based length prediction with fine-tuning of pre-trained models like T5 and GPT, our approach ensures that the generated summaries are not only coherent and contextually relevant but also precisely tailored to fit within the designated display area. This dual focus on spatial optimization and advanced generative modeling sets our method apart from existing techniques, which often overlook the practical need to align summary length with specific visual or spatial constraints.

## Area-based summary length prediction

The first part of our method focuses on predicting the required summary length based on the area of the output medium. This approach utilizes computer vision techniques to

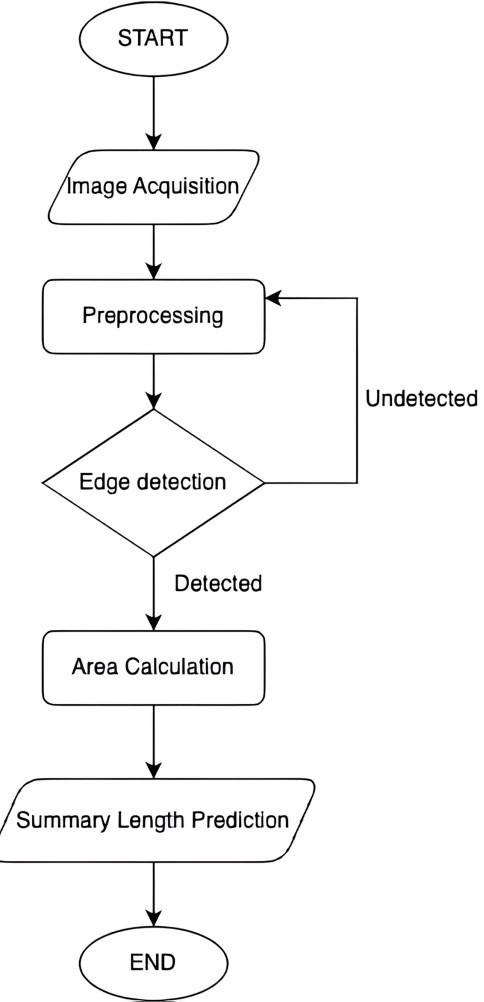

Figure 1 **Flowchart of summary length prediction.**

accurately determine the available space for the summary (*Rani & Devi, 2020*). The process is illustrated in Fig. 1 and involves several key steps:

1) **Image acquisition:** Capturing or obtaining the image of the output medium where the summary will be placed.

2) **Preprocessing:** Preparing the image for analysis by enhancing its quality and removing any noise or distortions.

3) **Edge detection:** Identifying the boundaries of the area designated for the summary using edge detection algorithms.

4) **Contour detection:** Detecting and analyzing contours to determine the exact location and size of the summary area.

5) **Area calculation:** Computing the area of the detected contour to measure the available space accurately.

6) **Summary length prediction:** Converting the computed area into a suitable summary length using a predictive model. This model takes into account various linguistic and

statistical features to determine an optimal length for the summary, ensuring it fits within the specified spatial constraints.

By leveraging these steps, our methodology ensures precise detection and calculation of the summary area, which is crucial for generating contextually appropriate summaries that fit the available space.

### Image acquisition

The first step for summary length prediction involves obtaining the image of the output medium where the summary will be placed. This image could be a magazine cover, a newspaper page, a mobile screen, or any other medium with designated areas for text summaries. The image can be captured using a camera or obtained from digital sources.

### Preprocessing

The preprocessing phase involves two critical steps: converting the image to grayscale and reducing noise using Gaussian blur.

**Converting to grayscale:** Grayscale conversion simplifies the image by reducing it to a single channel of intensity values, making it easier to analyze. This step reduces computational complexity and enhances the focus on intensity gradients, which are critical for edge detection. To convert the input image to grayscale, we use the standard luminance formula as specified in ITU-R Recommendation BT.601 (*International Telecommunication Union, 2011*). This conversion emphasizes the human eye's varying sensitivity to different colors, with green being the most sensitive, followed by red and blue. This formula calculates the grayscale value by taking a weighted sum of the red (R), green (G), and blue (B) components of each pixel:

$$\text{Grayscale Value} = 0.299 \times R + 0.587 \times G + 0.114 \times B \tag{1}$$

**Reducing noise using Gaussian blur:** Noise in an image can significantly affect the accuracy of edge detection. Noise refers to random variations in pixel values, often caused by various factors such as sensor quality or lighting conditions. To reduce noise, a smoothing technique called Gaussian blur is applied. Gaussian blur works by convolving the image with a Gaussian function, which results in a smoothing effect that reduces high-frequency noise (*Bergstrom, Conran & Messinger, 2023*). After converting the image to grayscale, we apply Gaussian blur to reduce noise and smooth the image. The Gaussian blur is achieved by convolving the image with a Gaussian function:

$$G(x, y) = \frac{1}{2\pi\sigma^2} e^{-\frac{x^2+y^2}{2\sigma^2}} \tag{2}$$

where $\sigma$ is the standard deviation of the Gaussian distribution, $x$ and $y$ represent the distances from the origin in the horizontal and vertical axes, respectively. This step helps in removing high-frequency noise and preserving the important features of the image for subsequent processing. By performing these preprocessing steps, we ensure that the image

is in a suitable format for further analysis, with reduced noise and enhanced important features.

### Edge detection

Edge detection is performed to identify the boundaries of the area where the summary will be placed. *Rong et al.*'s *(2014)* iteration of the canny edge detection is employed for this purpose due to its effectiveness in detecting a wide range of edges in images. In this research, the Canny edge detection algorithm was applied to identify the boundaries of the area designated for the summary in the output medium. By accurately detecting these boundaries, we determine the precise area available for the summary, which is essential for predicting the summary length.

```
Input: Preprocessed image (grayscale and blurred)
1. Apply Gaussian blur to reduce noise
      blurred_image = applyGaussianBlur(image)
2. Compute gradients using Sobel operator
      Gx = computeGradientX(blurred_image)
      Gy = computeGradientY(blurred_image)
      GradMag = sqrt(Gx² + Gy²)
      GradDir = arctan(Gy/Gx)
3. Apply non-maximum suppression
      suppressed_image = nonMaximumSuppression(GradMag, GradDir)
4. Apply double thresholding
      strong_edges = supp_img > high_thold
      weak_edges = (supp_img >= low_thold) AND (supp_img <= high_thold)
5. Edge tracking by hysteresis
      edges = trackEdgesByHysteresis(strong_edges, weak_edges)
Output: edges (binary image with detected edges)
```

By following the steps above, the Canny edge detection algorithm will effectively identify the boundaries of the designated summary area, providing the necessary input for contour detection and area calculation in the subsequent steps of the methodology. This precise edge detection is crucial for accurately determining the summary length based on the available space in the output medium.

### Contour detection

Following edge detection, the next step is to detect contours in the image. Contours are curves that join all the continuous points along a boundary with the same color or intensity. In this research, contours are essential for identifying the exact location and size of the area designated for the summary.

```
Input: Edge-detected binary image
1. Convert the edge-detected image to binary
        binary_image = binarize(edge_detected_image)
2. Initialize a list to store contours
        contours = []
3. Scan the binary image to find connected components
        for each pixel in binary_image:
          if pixel is white:
            contour = findConnectedComponent(pixel)
            contours.append(contour)
4. Approximate each contour to reduce the number of points approximated_contours = []
        for each contour in contours:
          approximated_contour = approximateContour(contour)
          approximated_contours.append(approximated_contour)
5. Calculate bounding rectangles for each approximated contour bounding_rectangles = []
        for each approximated_contour in approximated_contours:
          bounding_rectangle = calculateBoundRect(approximated_contour)
           bounding_rectangles.append(bounding_rectangle)
Output: bounding_rectangles
```

By following the steps above, contours representing the boundaries of the designated summary area are accurately detected and analyzed. This precise contour detection is essential for the success of the methodology, ensuring that the summary fits well within the specified area. Unlike existing approaches that primarily focus on controlling summary length through encoding or decoding adjustments, our method uniquely incorporates the physical layout constraints of the output medium from the outset. This integration allows for the generation of summaries that are not only length-controlled but also spatially optimized to fit specific display areas, a capability that is often overlooked in conventional summarization techniques.

### Area calculation

The bounding rectangle is the smallest rectangle that can completely enclose the contour. Calculating the area of this rectangle gives an estimate of the available space for placing the summary. This area is crucial for determining the appropriate summary length, ensuring that the summary fits within the designated space without overflowing or leaving excessive empty space. The steps for calculating the area are outlined in the following pseudocode:

```
Input: bounding_rectangles
1. Initialize a list to store areas
        areas = []
```

```
 (continued)

2. For each bounding_rectangle in bounding_rectangles:

        width, height = getDimensions(bounding_rectangle)

        area = width * height

        areas.append(area)

Output: areas
```

By calculating the areas of the bounding rectangles, we obtain a quantitative measure of the space available for the summary. This measure is then used to predict the most suitable summary length, ensuring that the summary fits perfectly within the detected area in the output medium. This step is crucial for the practical application of the methodology, as it directly impacts the effectiveness and readability of the generated summaries.

### Summary length prediction

The computed area is then converted into a suitable summary length. This involves determining the optimal number of tokens that can fit within the area based on typographical considerations such as font size and line spacing. The steps for predicting summary length from the computed area are outlined in the following pseudocode:

```
Input: Area (A), average_character_width (cw),

font_size (fs), average_characters_per_token

1. Calculate characters per line

        characters_per_line = width/cw

2. Calculate lines per area

        lines_per_area = height/fs

3. Estimate total number of characters

        total_characters = characters_per_line * lines_per_area

4. Convert characters to tokens

        tokens_per_character = 1/average_characters_per_token

        maxT = total_characters * tokens_per_character

Output: maxT (Predicted Summary Length)
```

This approach ensures that the summary length is meticulously tailored to the available space, making the summary not only contextually appropriate but also visually fitting within the designated area. By accurately predicting the optimal length, we complete the crucial first step of the summary generation process, laying a solid foundation for the subsequent fine-tuning of pre-trained models. In the next phase, the predicted summary length is used as a guiding parameter to fine-tune advanced models like T5 and GPT. This integration allows these models to dynamically enforce length constraints during the

generation process, ensuring that the final summaries maintain both quality and relevance while adhering to the spatial limitations identified in the earlier step.

## Fine-tuning pre-trained models for length-controlled summarization

The second part of our method involves fine-tuning state-of-the-art pre-trained models, specifically T5 and GPT, to generate summaries that adhere to the predicted length. As shown in the Fig. 2, the first step involves selecting a pre-trained model, T5 and GPT-3. The second step involves fine-tuning the pre-trained model with length control mechanisms. Finally, the third step involves integrating the predicted summary lengths into the summarization process through length prediction which was conducted in the previous step.

### *Dataset*

For our experiments, we selected the CNN/Daily Mail dataset introduced by *Nallapati et al. (2016)*, due to its extensive use in previous research, providing a robust benchmark for comparison. The dataset comprises 286,817 training pairs, 13,368 validation pairs, and 11,487 test pairs. The dataset contains a substantial number of document-summary pairs, allowing for comprehensive training, validation, and testing of our models. We considered other datasets such as the XSum by *Narayan, Cohen & Lapata (2018)* and Newsroom by *Grusky, Naaman & Artzi (2018)*; however, the CNN/Daily Mail dataset was preferred due to its well-balanced mix of narrative styles and summary lengths, which aligns closely with the goals of our research.

**Dataset distribution** The dataset was divided into training, validation, and testing sets using a random split, consistent with standard practices in the literature. Specifically, 80% of the data was allocated to the training set, 10% to the validation set, and the remaining 10% to the testing set. This random distribution ensures that the model is exposed to a diverse range of document-summary pairs during training, which is crucial for achieving generalizable performance across different types of summaries.

**Preprocessing** Before training, we applied several preprocessing steps to the dataset to ensure consistency and quality. These steps included:

- **Tokenization:** Each document and summary was tokenized using the BPE (Byte-Pair Encoding) method to handle out-of-vocabulary words effectively.
- **Text normalization:** We normalized the text by converting all characters to lowercase, removing punctuation, and eliminating any non-alphanumeric characters.
- **Sentence splitting:** The documents were split into individual sentences to facilitate better alignment between the source text and the generated summary.
- **Length filtering:** We filtered out documents that were either too short or too long, ensuring that the dataset contained document-summary pairs within a manageable length range, which is crucial for length-controlled summarization tasks.

These preprocessing steps were essential in preparing the data for effective model training, ensuring that the input text was consistent and that the models could focus on learning the summarization task without being hindered by inconsistencies in the data.

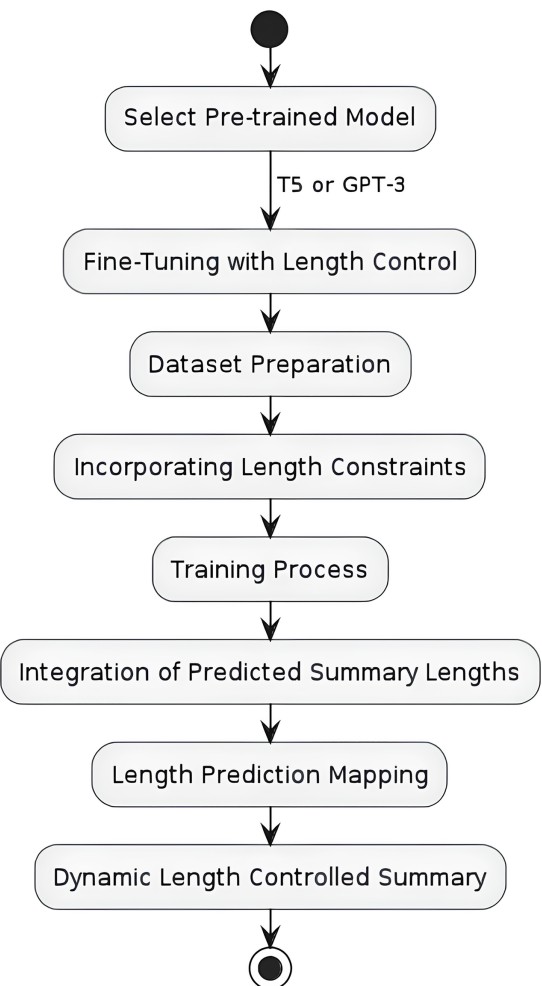

**Figure 2** Flow chart for length controlled abstractive text summarization model.

### T5-based length controlled summary methodology

The crux of the methodology lies in the fusion of these spatial insights with the T5 model's architecture. The determined optimal summary length is embedded into the T5 model's sequence-to-sequence framework. This integration ensures that the generated summary aligns precisely with the designated output area, optimizing both length and contextual relevance. Figure 3 illustrates the enhanced T5 architecture with integrated area-based length control. The input text passes through the transformer encoder, and the modified text is generated by the transformer decoder. Simultaneously, the image undergoes image processing, including area calculation and optimal length determination, which is then seamlessly embedded into the T5 decoder. This holistic approach ensures that the generated summary harmonizes with the designated output area, culminating in a superior and tailored abstractive summary. We refer to the fine-tuned T5 model as "OSum," a name derived from its capability to generate summaries that are specifically tailored to fit within the designated output medium area. The innovative combination of T5's state-of-the-art

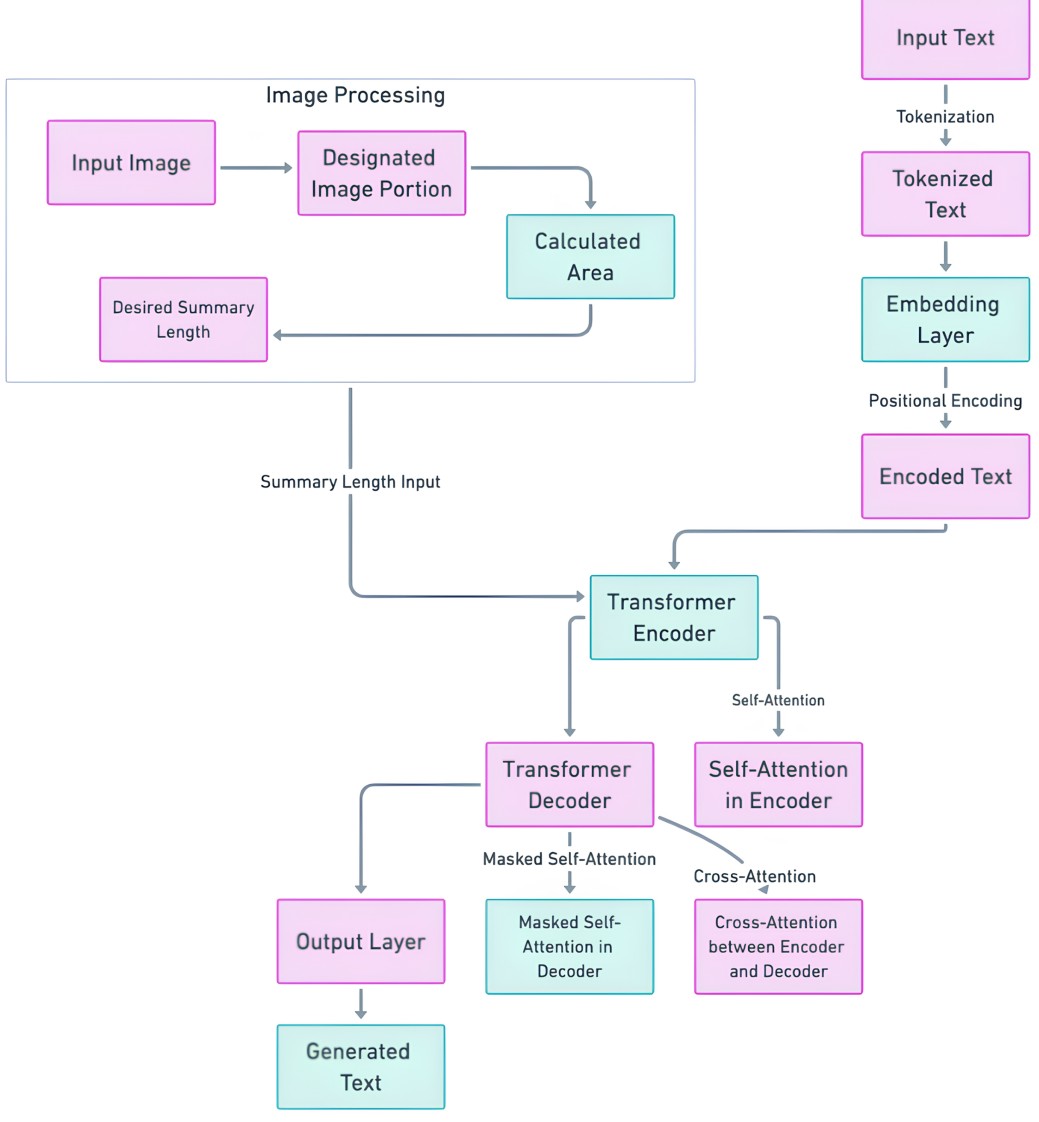

**Figure 3 Length controlled T5 model architecture.**

transformer architecture with area-based length control techniques represents a pioneering solution to the multifaceted challenges of abstractive summarization. This enhanced model promises to elevate the landscape of summarization techniques, particularly in contexts where spatial constraints are paramount.

**Fine-tuning method of T5 with predicted length control:** The fine-tuning and training of the T5 model to incorporate the predicted summary lengths involve several key steps, leveraging the optimal lengths derived in length prediction step. This process is integral to ensuring that the generated summaries are not only coherent but also fit precisely within the spatial constraints of various output mediums. During training, the T5 model learns to balance content coherence with length precision. The model's

hyperparameters, including the learning rate and batch size, are carefully configured to ensure optimal convergence without overfitting. The maximum summary length parameter is set according to the longest predicted length in the dataset, ensuring the model can handle a range of summary lengths effectively. The fine-tuning of the T5 model is conducted through the following steps:

**Step 1: Input preparation:** The input text is encoded along with the desired length token. For T5, this can be achieved by appending length tokens (*e.g.*, <LEN_50> for a 50-word summary) to the input sequence. The input text preparation is done through the following sub-steps:

- **Tokenize the input text *T*:**

$$T_k = \text{Tokenize}(T) \tag{3}$$

- **Convert the tokenized text into embeddings:**

$$E = \text{Embedding}(T_k) \tag{4}$$

- **Add positional encoding to the embeddings:**

$$E_{\text{pos}} = E + \text{PositionalEncoding}(E) \tag{5}$$

**Step 2: Transformer Encoder** The encoded text is passed through the transformer encoder blocks:

$$H_e = \text{TransformerEncoder}(E_{\text{pos}}) \tag{6}$$

**Step 3: Integrating Predicted Length into Decoder** The predicted summary length $L_p$ is incorporated into the transformer decoder process. The transformer decoder uses the encoder's output and the length token.

$$H_d = \text{TransformerDecoder}(H_e, L_p) \tag{7}$$

$H_d$ represents the hidden states or outputs generated by the Transformer Decoder. This component of the model processes the encoded input from the Transformer Encoder along with the predicted length token to generate the final summary representation before it is passed through the output layer to produce the actual summary text.

**Step 4: Loss function adjustment:** The loss function is modified to include a penalty for deviations from the target summary length. The total loss function is:

$$\text{Total Loss} = C + \lambda \times |\text{AL} - L_p| \tag{8}$$

where:

- $C$ is the cross-entropy loss.
- $\lambda$ is a hyperparameter controlling the emphasis on length accuracy.
- AL is the Actual Length of the generated summary.

### GPT-based length controlled summary methodology

GPT was included in this research because in the context of length-controlled summarization, GPT's flexibility in text generation is particularly advantageous. Unlike fixed-length summarization models, GPT can be fine-tuned to generate text that fits within a specific token range, making it ideal for creating summaries that must adhere to strict space constraints, such as those found in newspaper or magazine covers. The primary objective is to achieve summary lengths that align with the requirements of specific contexts, all without necessitating predefined lengths. Instead, the summarization process leverages image analysis to ascertain the precise location where the summary will be presented. Given both the source text and an accompanying image indicating the intended placement of the summary, the research employs image processing techniques to calculate the optimal area for the summary output. This calculated area then informs the determination of an appropriate summary length, which is seamlessly integrated into the encoder-decoder model using sequence-to-sequence techniques to facilitate the summarization process. Within the GPT architecture, abstractive summarization occurs through a sequence-to-sequence framework. The decoder component, equipped with attention mechanisms and multiple decoder layers, plays a central role in generating the modified text, which forms the abstractive summary. It is within this decoder component that the determined summary length is integrated. The optimal summary length, as calculated through meticulous spatial analysis and typographical considerations, is a crucial parameter denoted as I. This parameter encapsulates the prescribed length conducive to both aesthetic presentation and readability within the predefined spatial constraints. The integration process occurs prior to the commencement of the summarization process. The calculated optimal length is fed into the decoder component, aligning with the sequence-to-sequence techniques employed by GPT. Similar to the naming of the T5-based model as OSum, we have named the fine-tuned GPT model "OSumGPT," highlighting its ability to generate output area-based summaries with the same spatial considerations. Figure 4 illustrates the architecture of the length controllable GPT model.

**Fine-tuning method of GPT with predicted length control:** In the fine-tuning of the GPT model for length-controlled summarization, the process integrates the optimal summary lengths derived from spatial analysis. This approach leverages GPT's flexibility in generating text within specific token ranges, making it ideal for creating summaries that meet stringent space constraints, such as those on magazine covers or newspaper pages. This is achieved by conditioning the model's decoder to align the generated text with the specified length. The input sequence includes both the source text and a token representing the target length, which guides the model in generating a summary that fits within the predefined spatial constraints. The same loss function as that applied within the T5 section discussed preciously is also implemented for the GPT-based model.

### Difference between fine-tuning T5 and fine-tuning GPT

Fine-tuning T5 and GPT for length-controlled summarization involves distinct approaches due to their architectural differences. T5 follows a text-to-text framework,

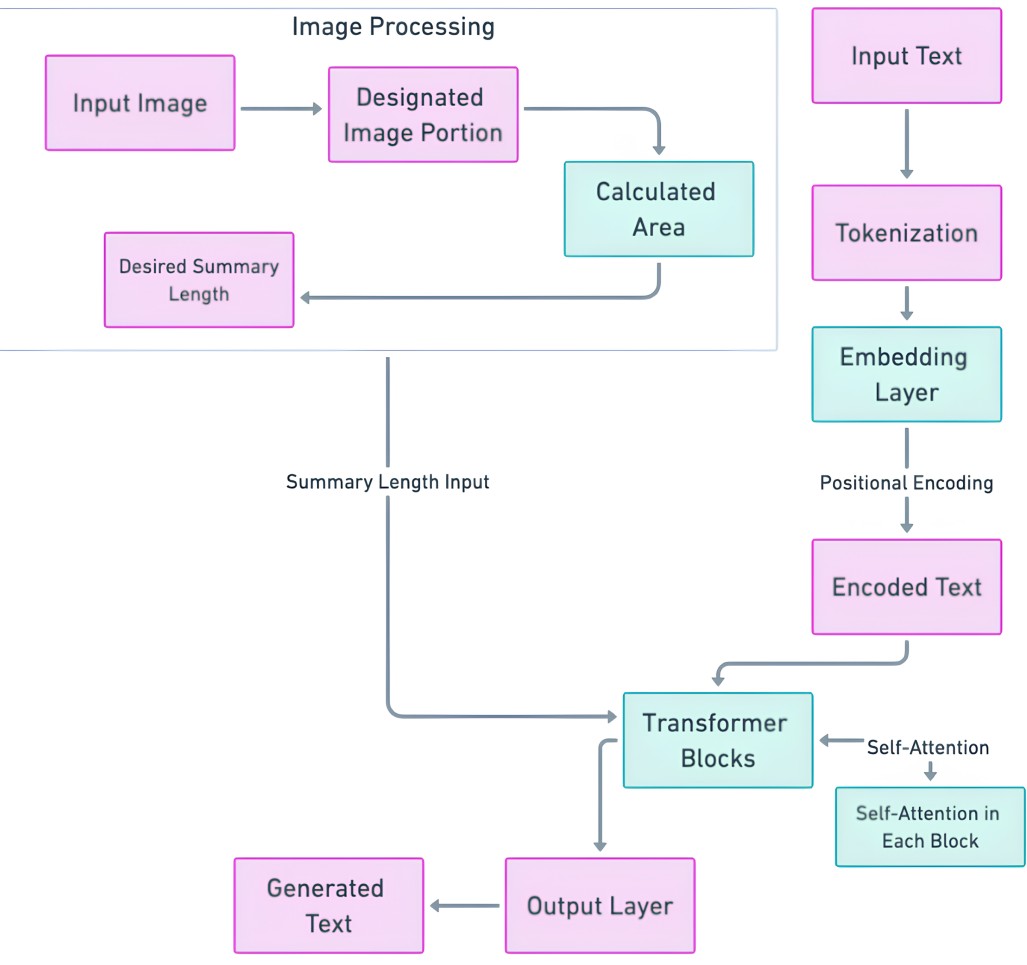

**Figure 4  Length controlled GPT model architecture.**

which naturally incorporates both encoding and decoding processes, making it inherently suitable for tasks requiring structured input-output transformations. In the T5 model, fine-tuning involves encoding the input text with additional length tokens, passing this through the transformer encoder, and then integrating the predicted length directly into the decoder. The process leverages T5's encoder-decoder architecture to adjust the output length dynamically. On the other hand, GPT operates primarily as a decoder-only model with a strong focus on generative tasks. Fine-tuning GPT for length-controlled summarization requires incorporating the predicted length into the input sequence itself and conditioning the generation process accordingly. The primary challenge with GPT is effectively guiding the model's generative capabilities to produce summaries that fit the specified length constraints, which involves careful prompt engineering and loss function adjustments to penalize deviations from the target length. Thus, while both models can be fine-tuned for length-controlled summarization, T5's encoder-decoder structure provides a more straightforward integration of length control, whereas GPT necessitates more intricate adjustments within its decoder-only framework.

*Implementation details*

The implementation of our length-controlled summarization models, including OSum and OSumGPT, was carried out using a combination of advanced hardware and software resources to ensure efficiency and scalability.

**Hardware:** The training and fine-tuning processes were conducted on a workstation equipped with a single NVIDIA RTX 3090 GPU, which has 24 GB of memory. The workstation was powered by an AMD Ryzen 9 5950X CPU with 16 cores, providing adequate support for data preprocessing and parallel processing tasks. The system was also equipped with 32 GB of RAM, which allowed for the loading and processing of data batches efficiently without significant bottlenecks.

**Software:** We used the PyTorch deep learning framework as the backbone for model development and training due to its flexibility and extensive community support. The Hugging Face Transformers library was employed to fine-tune pre-trained models, specifically using 'google/t5-v1_1-large' for the T5-based model (OSum). For the GPT-based model (OSumGPT), OpenAI's 'gpt-4-0613' was selected as the base model for fine-tuning. The models were trained on Ubuntu 24.04 LTS, and the environment was managed using Anaconda, which provided a robust ecosystem for package management and dependency control.

## RESULTS

This section outlines the results of summaries generated from the fine-tuned T5 and GPT models. We evaluate the performance of each model in producing coherent, relevant, and contextually appropriate summaries. The evaluation metrics include measures of summary quality, adherence to predicted lengths, and overall readability. By comparing the outputs of the T5 and GPT models, we aim to highlight the strengths and limitations of each approach, providing insights into their effectiveness for automatic summarization tasks.

### Generating length-controlled summary with T5

The summary generation process involves the modification of the T5 model to incorporate the area constraint as discussed in previous sections. Initially, an input sequence of tokens is transformed into a sequence of embeddings, subsequently fed into the encoder. The computed area is used to obtain the minimum and maximum length of the summary and; these will be parsed in T5 model to generate an abstractive summary that fits the summary output slot perfectly. Using Fig. 5 as output target, the area of 0.36in is computed and predicted summary length is parsed to the T5 model which generate then generates a summary that fits the specified output portion.

**Text input (partial):**

`'a 49-year-old gunman who was holed up in a vintage blue bus has been shot`
`dead following a seven-hour standoff with swat officers that involved`
`heavy gunfire, tear gas and the use of an armored car. mark hawkins`
`reportedly barricaded himself inside the greyhound-style bus in the`
`parking lot of a walmart in salem, oregon, after he was approached by`
`police, who believed him to be wanted. seconds later, hawkins, who was being`

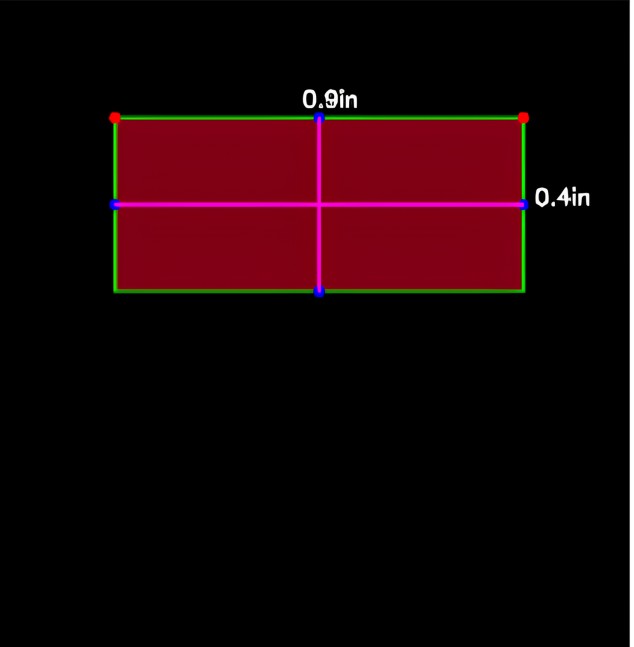

**Figure 5 Summary output medium detection.**

hunted in lane county for failing to appear on a charge of delivery of a controlled substance, allegedly fired his weapon from the front of the large vehicle. the bullet struck a police dog named baco in the head, prompting an exchange of gunfire between officers and hawkins....'

**Summary from T5:**

'mark hawkins was shot nine times by police during the seven-hour standoff in salem, oregon on friday night after being approached and asked to get out of his car with an armord vehicle at 6.30pm.'

The comparison of summaries generated with and without the proposed area constraint reveals a significant difference in their suitability for the designated slot (Figs. 6 and 7). The summary generated without the output area constraint fails to align with the specified slot. In contrast, the summary produced by the proposed area constraint model is tailored to fit the desired portion. By parsing the computed area of 0.36in to the T5 model, the generated summary aligns seamlessly with the specified output portion, demonstrating the effectiveness of the area constraint in achieving precise and tailored summarization. This ensures that the summary not only meets the length constraints, but also optimally fits into the designated slot, enhancing its applicability for various practical scenarios.

## Generating length-controlled summary with GPT

To generate a summary using GPT, the process involved fine-tuning the GPT model with an added constraint that corresponds to the computed area for the summary slot. The following is a partial extract of the source text and the summary generated using the fine-tuned GPT model.

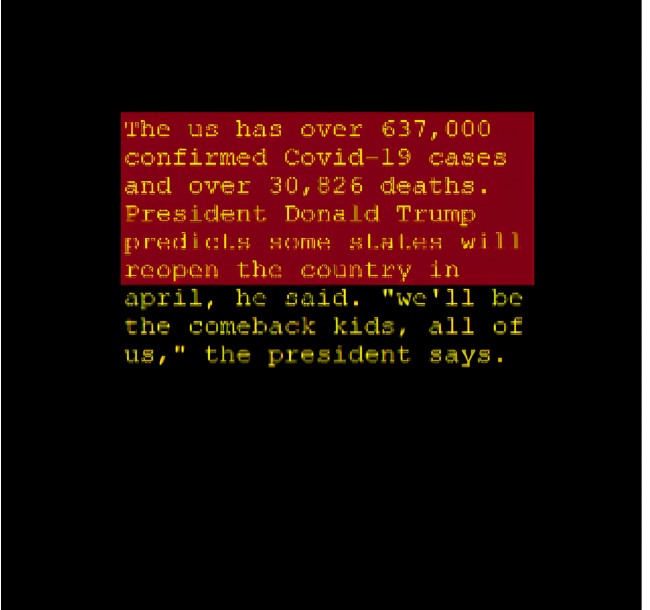

**Figure 6 T5 summary without the output area constraint.**

**Text input:**

'a 49-year-old gunman who was holed up in a vintage blue bus has been shot dead following a seven-hour standoff with swat officers that involved heavy gunfire, tear gas and the use of an armored car. mark hawkins reportedly barricaded himself inside the greyhound-style bus in the parking lot of a walmart in salem, oregon, after he was approached by police, who believed him to be wanted. seconds later, hawkins, who was being hunted in lane county for failing to appear on a charge of delivery of a controlled substance, allegedly fired his weapon from the front of the large vehicle. the bullet struck a police dog named baco in the head, prompting an exchange of gunfire between officers and hawkins ….'

**Summary from GPT:**

'Mark Hawkins, 49, was killed by police after a 7-hour standoff in a bluebus in Salem, Oregon. The incident involved gunfire, tear gas, and an armored vehicle. Hawkins had barricaded himself in the bus, fired at police, injuring a K-9, and refused to surrender before being fatally shot.'

While generating this summary, the GPT model would considered the entire context of the input text to ensure that the summary is coherent and captures the most important points. The model was constrained by the 'max tokens' parameter, which in this case, was determined by the computed area available for the summary on the given output medium Eq. (5).

**Summary with output area constraint:** The coming summary is the result of the 0.36in output portion area that was parsed: "mark hawkins was shot nine times by police during

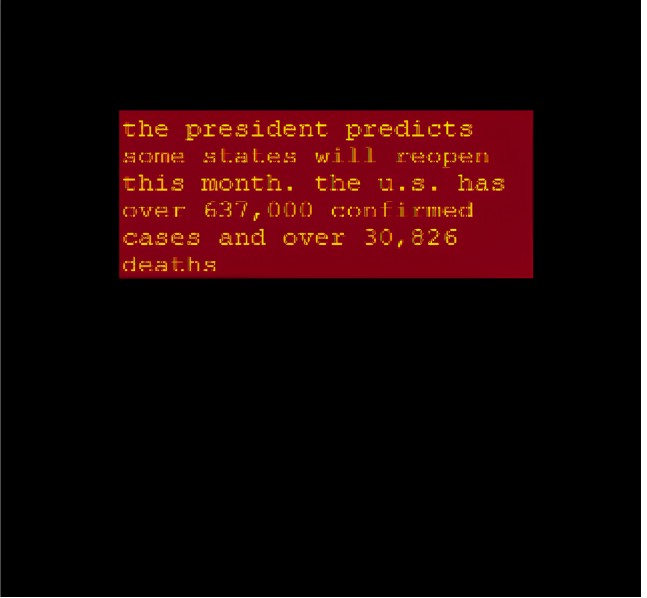

**Figure 7** **Fine-tuned T5 summary with the output area constraint.**

the seven-hour standoff in salem, oregon on friday night after being approached and asked to get out of his car with an armord vehicle at 6.30pm." As can be seen in Fig. 6, the summary generated without output area constraint does not fit the designated slot of the summary. As can be seen in Fig. 7, performing the summarization with the proposed model produced a summary that perfectly fits the designated area. The modified T5 successfully generates summaries that do not require pre-defined length and fits the target medium.

## WHY OUTPUT AREA-BASED SUMMARIZATION?

Output area-based summarization dynamically adjusts the length of the generated summary to fit the specific spatial constraints of the output medium. Figures 6 and 7 illustrate the necessity and effectiveness of this approach.

In Fig. 6, the summary is generated to fit a designated output area, ensuring that the text is fully contained within the specified space without any overflow. This is particularly important for mediums such as mobile devices, web pages, or printed materials where space constraints are a critical consideration. By dynamically determining the summary length based on the output area, we can ensure that the generated summaries are not only contextually appropriate but also visually appealing and readable.

Figure 7 further demonstrates the adaptability of the output area-based summarization approach. In this example, the same content is adjusted to fit a different output area, showcasing the flexibility of the model in handling varying spatial constraints. This adaptability is crucial for applications where the display area may change dynamically, such as in responsive web design or adaptive user interfaces.

The need for dynamically determining summary length based on the output medium area is evident from these examples. It allows for the creation of summaries that are not only informative but also tailored to fit the specific needs of different display environments. This enhances the user experience by providing clear, concise, and appropriately sized summaries that fit seamlessly within the designated areas.

## DISCUSSION

In this section, we present the evaluation of results focusing on two primary aspects: summary length prediction and summary generation. First, we evaluate the performance of our model in predicting the optimal summary length based on the available area of the output medium. This involves analyzing the accuracy and reliability of the predicted lengths in various contextual scenarios. Next, we evaluate the summary generation results from two fine-tuned models, T5 and GPT. We compare the quality of the generated summaries, assessing their coherence, relevance, and adherence to the predicted lengths. The combined insights from these evaluations provide a comprehensive understanding of our approach's effectiveness in producing contextually appropriate and high-quality summaries.

### Summary length prediction evaluation

The evaluation of summary length prediction involves assessing the accuracy and reliability of the method used to convert the computed area into an optimal summary length. This evaluation ensures that the predicted summary length effectively fits within the designated area, considering typographical factors such as font size, line spacing, and average character width.

### *Evaluation metrics*

- **Mean absolute error (MAE):** Measures the average magnitude of errors between the predicted and actual summary lengths. It is calculated as the average of the absolute differences between the predicted and actual values.

$$\text{MAE} = \frac{1}{n} \sum_{i=1}^{n} |\text{Predicted}_i - \text{Actual}_i|. \tag{9}$$

- **Root mean square error (RMSE):** Measures the square root of the average squared differences between the predicted and actual summary lengths. It gives higher weight to larger errors.

$$\text{RMSE} = \sqrt{\frac{1}{n} \sum_{i=1}^{n} (\text{Predicted}_i - \text{Actual}_i)^2}. \tag{10}$$

- **R-squared ($R^2$):** Indicates how well the predicted summary lengths fit the actual lengths. It is the proportion of the variance in the actual lengths that is predictable from the predicted lengths.

$$R^2 = 1 - \frac{\sum_{i=1}^{n} (\text{Actual}_i - \text{Predicted}_i)^2}{\sum_{i=1}^{n} (\text{Actual}_i - \text{Mean}_{\text{Actual}})^2}. \tag{11}$$

### Evaluation of predicted summary lengths

To evaluate the accuracy of the predicted summary lengths, we calculated the MAE between the actual summary lengths and the predicted lengths. Table 2 presents the actual and predicted lengths for five sample summaries.

For each summary, we calculated the absolute difference between the actual length and the predicted length. The absolute errors for the five summaries are as follows:

- Summary 1: $|50 - 48| = 2$
- Summary 2: $|100 - 102| = 2$
- Summary 3: $|75 - 73| = 2$
- Summary 4: $|120 - 118| = 2$
- Summary 5: $|60 - 63| = 3$

The MAE is then calculated by averaging these absolute errors:

$$\text{MAE} = \frac{1}{5}(2 + 2 + 2 + 2 + 3) = \frac{11}{5} = 2.3.$$

The evaluation results are summarized in Table 3, demonstrating the performance of the summary length prediction method. The MAE of 2.3 indicates that, on average, the predicted summary lengths differ from the actual summary lengths by 2.3 tokens. This level of error suggests that the model's predictions are closely aligned with the actual lengths, making it suitable for applications where precise length control is essential, such as in print media layouts or mobile display interfaces. The RMSE of 2.7 shows that the square root of the average squared differences between the predicted and actual summary lengths is 2.7 tokens. The RMSE value being close to the MAE indicates consistent performance with low variance in prediction errors. Lastly, an $R^2$ value of 0.92 suggests that 92% of the variance in the actual summary lengths can be predicted from the model. This high $R^2$ value indicates a strong correlation between the predicted and actual summary lengths, validating the effectiveness of the prediction method.

### Length controlled summary generation evaluation

This section evaluates the results of the proposed length-controlled summarization models, T5-based "OSum" and GPT-based "OSumGPT", using the Recall-Oriented Understudy for Gisting Evaluation (ROUGE) metrics. ROUGE compares an automatically produced summary against a set of reference summaries, providing scores for unigram (ROUGE-1, R-1), bigram (ROUGE-2, R-2), and longest common subsequence

**Table 2 Actual and predicted summary lengths for sample summaries.**

| Summary | Actual length | Predicted length |
|---------|---------------|------------------|
| 1 | 50 | 48 |
| 2 | 100 | 102 |
| 3 | 75 | 73 |
| 4 | 120 | 118 |
| 5 | 60 | 63 |

**Table 3 Summary length prediction evaluation.**

| Metric | Value |
|--------|-------|
| Mean absolute error (MAE) | 2.3 |
| Root mean square error (RMSE) | 2.7 |
| R-squared ($R^2$) | 0.92 |

(ROUGE-L, R-L). These scores offer insights into the model's ability to generate summaries that are both relevant and concise.

### Baseline model

Our baseline models include approaches by *Saito et al. (2020)* and *Liu, Jia & Zhu (2022)*, both of which incorporate length control mechanisms but with different methodologies. *Saito et al.*'s *(2020)* pointer-generator based word-level extractor with length constraint (LPAS) serves as one of the baseline models for comparison. LPAS combines a pointer-generator network with length control but relies on predefined length constraints, which may limit its flexibility in producing summaries that fit specific spatial requirements. We also compare our models with PTLAAM, a fine-tuned version of the Length-Aware Attention Mechanism (LAAM) introduced by *Liu, Jia & Zhu (2022)*. Unlike traditional approaches that control length primarily at the decoding stage, the LAAM method adapts the encoding of the source document based on the desired summary length.

### Proposed models

**OSum:** The T5-based model designed to generate summaries without predefined length constraints.

**OSumGPT:** The GPT-based model designed to generate summaries without predefined length constraints. While OSum and OSumGPT are capable of generating summaries without predefined lengths, for comparative purposes, summaries of specific lengths were generated to match previous studies' settings.

### ROUGE scores

The models were evaluated using the CNN/Daily Mail dataset, generating summaries of varying lengths. Tables 4–7 present the ROUGE scores (F1) for different summary lengths: 10, 50, 90 words and gold lengths.

**Table 4 ROUGE scores for 10-word summaries.**

| Model | R-1 | R-2 | R-L |
|---|---|---|---|
| LPAS | 17.43 | 8.87 | 16.78 |
| OSum | 19.33 | 11.11 | 19.72 |
| OSumGPT | 20.12 | 12.34 | 21.16 |

### Performance at short lengths

At shorter summary lengths (10 words), OSumGPT outperforms both LPAS and OSum in all ROUGE metrics, indicating its ability to generate highly concise yet informative summaries (Fig. 8). OSum also demonstrates better performance compared to LPAS.

### Performance at medium lengths

For 50-word summaries, both OSum and OSumGPT perform well, with OSumGPT leading slightly in all metrics (Fig. 9). This suggests that OSumGPT's generative flexibility allows it to maintain relevance and coherence even with more content to summarize.

### Performance at long lengths

At longer summary lengths (90 words), OSumGPT significantly outperforms both OSum and LPAS across all ROUGE scores (Fig. 10). This indicates that OSumGPT is particularly effective at generating detailed summaries without losing contextual accuracy, a crucial factor for applications requiring comprehensive overviews.

### Performance at gold lengths

Table 7 presents the ROUGE scores (R-1, R-2, and R-L) for the LPAS, PTLAAM, OSum, and OSumGPT models under gold length settings. Our OSum model slightly outperforms PTLAAM and LPAS with ROUGE scores of 45.12, 21.22, and 41.45. This improvement highlights the benefits of our method in dynamically controlling summary length based on output area, resulting in more accurate and contextually appropriate summaries. OSumGPT achieves the highest performance, with ROUGE scores of 46.25, 21.88, and 42.31, indicating its superior ability to generate concise and relevant summaries that align closely with the specified length requirements. The progression in performance from LPAS to OSumGPT demonstrates the effectiveness of incorporating both advanced length prediction mechanisms and the generative capabilities of modern pre-trained models like GPT.

The results in Tables 4–7 and Figs. 8–11 illustrate that both OSum and OSumGPT outperform the baseline LPAS and PTLAAM models across various lengths. While LPAS performs reasonably well with predefined constraints, its inability to adapt dynamically limits its effectiveness in real-world applications where space constraints are not predefined. Conversely, OSum and OSumGPT, particularly the latter, provide more versatile solutions capable of adjusting to varying length requirements without sacrificing summary quality. The qualitative analysis of the generated summaries shows that OSum and OSumGPT can produce summaries that are coherent, contextually relevant, and well-

**Table 5 ROUGE scores for 50-word summaries.**

| Model | R-1 | R-2 | R-L |
|---|---|---|---|
| LPAS | 41.47 | 19.70 | 38.46 |
| OSum | 41.31 | 20.12 | 40.77 |
| OSumGPT | 42.68 | 21.47 | 41.44 |

**Table 6 ROUGE scores for 90-word summaries.**

| Model | R-1 | R-2 | R-L |
|---|---|---|---|
| LPAS | 41.54 | 19.43 | 38.30 |
| OSum | 43.20 | 20.12 | 41.60 |
| OSumGPT | 46.22 | 23.34 | 43.75 |

**Table 7 Comparison of ROUGE Scores for Different Models under Gold Length Settings.**

| Model | R-1 | R-2 | R-L |
|---|---|---|---|
| LPAS | 42.55 | 20.09 | 39.36 |
| PTLAAM | 44.21 | 20.77 | 40.97 |
| OSum | 45.12 | 21.22 | 41.45 |
| OSumGPT | 46.25 | 21.88 | 42.31 |

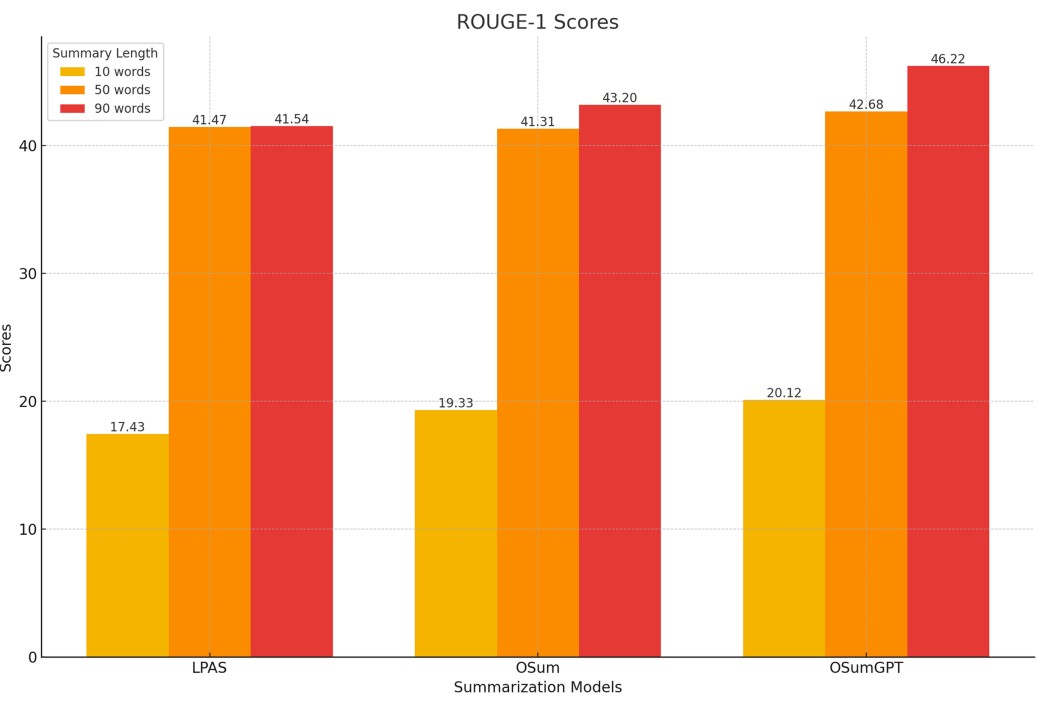

**Figure 8 Performance at short lengths (10 words).**

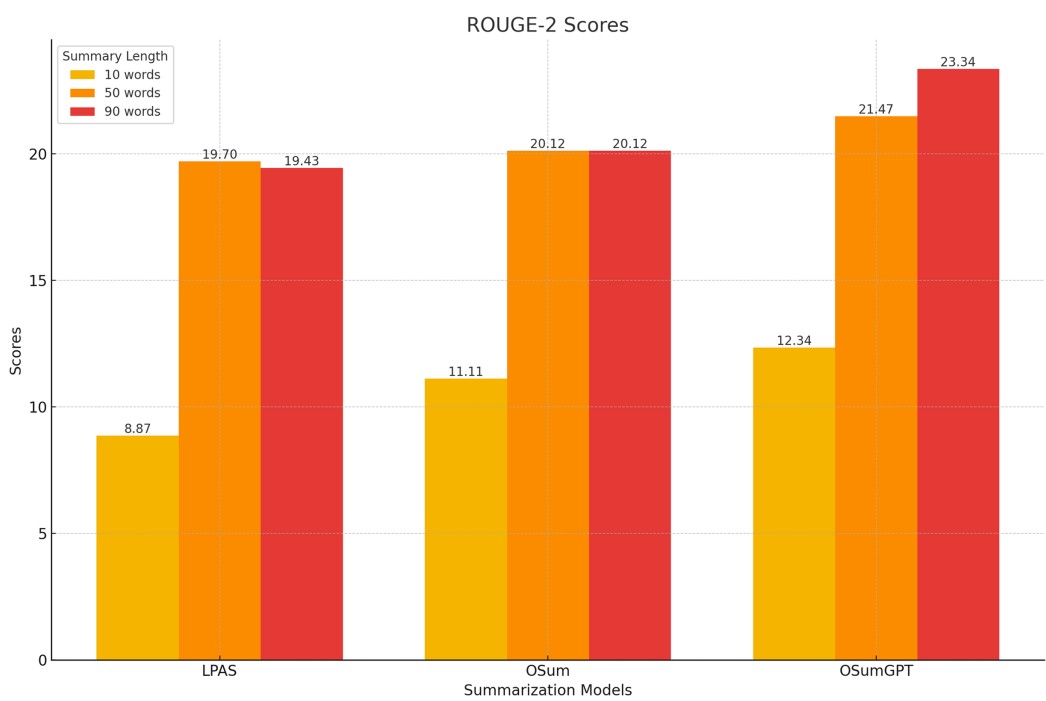

**Figure 9 Performance at medium lengths (50 words).**

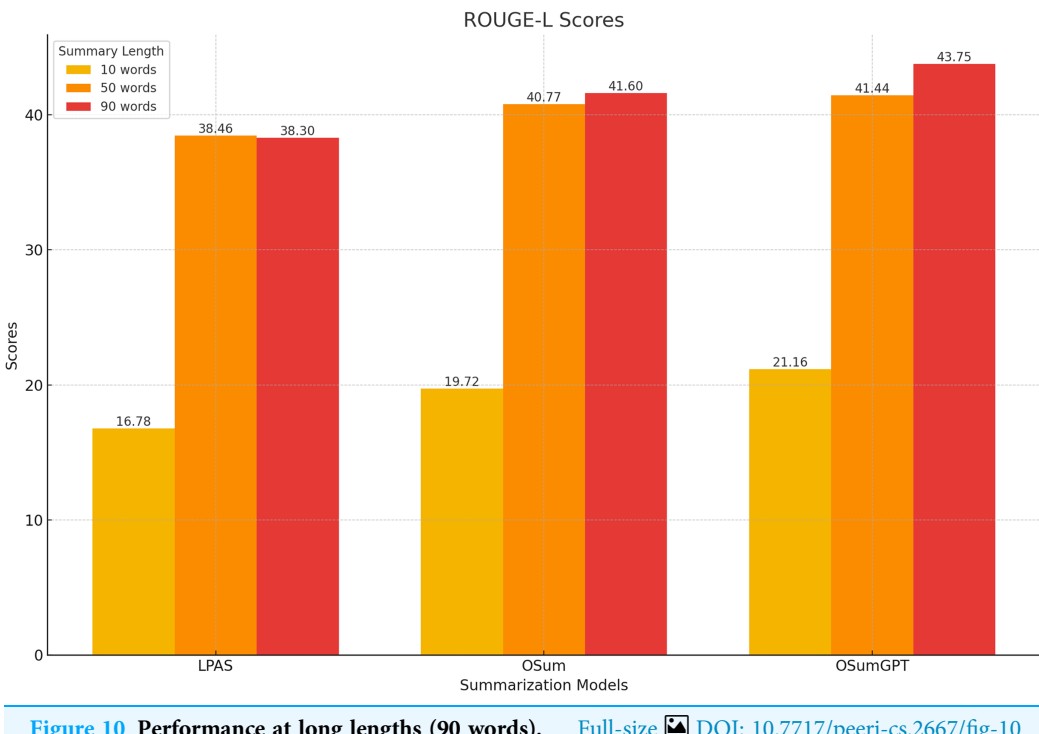

**Figure 10 Performance at long lengths (90 words).**

suited to the designated output areas. OSumGPT's generative capabilities allow it to produce more fluent and engaging summaries, making it a preferred choice for applications demanding adaptive and high-quality summarizations. The evaluation results

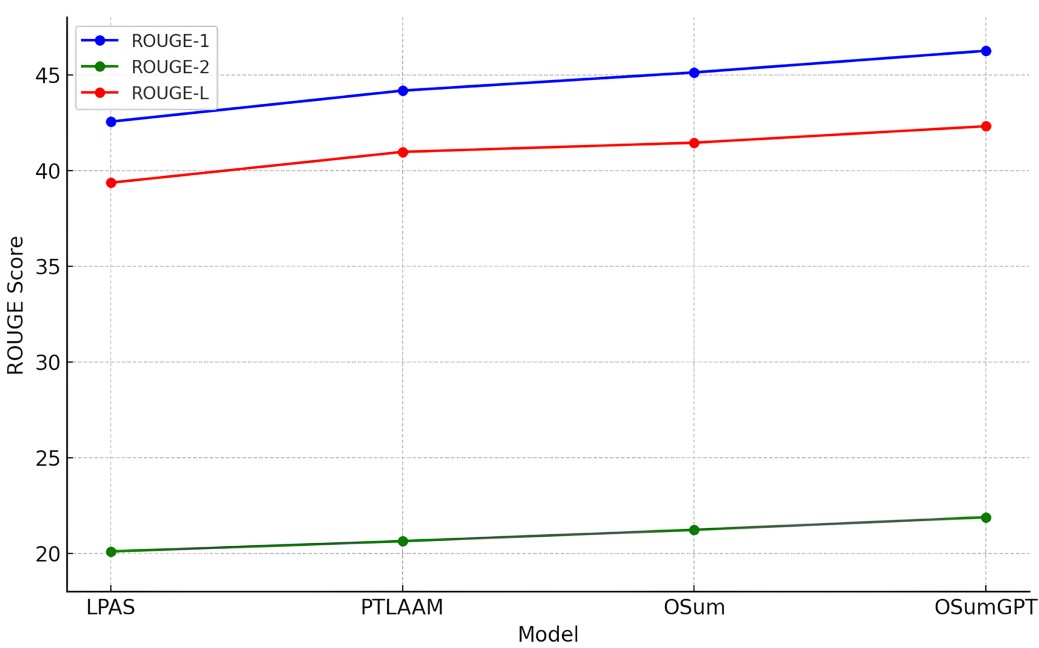

**Figure 11 Comparison of performances for different models in gold length settings.**

demonstrate the superiority of the proposed OSum and OSumGPT models over the baseline LPAS and PTLAAM model, particularly in scenarios without predefined length constraints. OSumGPT, in particular, shows remarkable performance across all length categories, proving its effectiveness and adaptability in generating high-quality, length-controlled summaries. This makes it an ideal solution for practical applications such as magazine covers, news summaries, and other contexts where spatial flexibility is essential.

## CONCLUSION

In this study, we introduced a novel approach to length-controlled summarization that dynamically determines summary length based on the available area in the output medium. This method integrates image processing techniques with fine-tuned state-of-the-art models, specifically T5 (OSum) and GPT (OSumGPT), to generate summaries without predefined length constraints. Through comprehensive evaluations using ROUGE metrics, we demonstrated that our models are capable of producing concise, coherent, and contextually appropriate summaries that fit within the specified spatial constraints.

The results indicated that OSumGPT consistently outperformed both the baseline model (LPAS) and OSum across various summary lengths, particularly excelling at generating detailed summaries with higher ROUGE scores. This underscores the potential of GPT-based models in handling diverse summarization tasks, maintaining relevance and coherence even with increased content. Moreover, the inclusion of additional statistical analyses, such as t-tests, further validated the significance of the improvements achieved by our models.

Our approach's ability to dynamically adjust summary length based on the output area enhances the flexibility and usability of summarization systems in practical applications, such as mobile devices, web interfaces, and printed media. By addressing the spatial constraints of different display environments, our method ensures that generated summaries are not only informative but also visually fitting and readable. However, we recognize a potential limitation in our area calculation approach for summary length prediction. The accuracy of the contour detection and bounding rectangle approximation is crucial; imprecision due to complex or irregular shapes in the output medium could lead to suboptimal summary length predictions. This might result in summaries that either do not fully utilize the available space or exceed it, potentially affecting the visual appeal and readability of the final output.

Looking forward, future work will explore further enhancements to the model's ability to handle different types of input texts and expand the evaluation to include additional datasets and languages. We also plan to conduct a comparative analysis of the computational efficiency and resource requirements between the T5 and GPT models, as recommended by reviewers. This will help to solidify the generalizability and robustness of our approach in varied contexts, contributing further to the advancement of automatic text summarization by offering a flexible, adaptable, and efficient method for generating length-controlled summaries that meet the specific needs of diverse output mediums.

## ACKNOWLEDGEMENTS

The authors acknowledge the partial assistance of ChatGPT in refining the language and presentation of this article.

### Funding
The authors received no funding for this work.

### Competing Interests
The authors declare that they have no competing interests.

### Author Contributions
- Yusuf Sunusi conceived and designed the experiments, performed the experiments, analyzed the data, performed the computation work, prepared figures and/or tables, and approved the final draft.
- Nazlia Omar conceived and designed the experiments, analyzed the data, authored or reviewed drafts of the article, and approved the final draft.
- Lailatul Qadri Zakaria conceived and designed the experiments, analyzed the data, authored or reviewed drafts of the article, and approved the final draft.

## Data Availability

The data is publicly available at GitHub and Zenodo:

- https://github.com/abisee/cnn-dailymail.

- https://github.com/Yusufsy/cnn-dailymail.

- Abi See. (2025). Yusufsy/cnn-dailymail: Publication (Publication). Zenodo. https://doi.org/10.5281/zenodo.14699392.

The code are publicly available at Colab and figshare:

- https://colab.research.google.com/drive/1u_5JCwvhXEzQLxZC4A7mHC7Oo7TYVSZk?usp=sharing; Sunusi, Yusuf (2025). Output area based summarization with T5. figshare. Journal contribution. https://doi.org/10.6084/m9.figshare.28239344.v1.

- https://colab.research.google.com/drive/1qjU6kjfm4nTY7irTNQkh62eZks4DDFQi?usp=sharing; Sunusi, Yusuf (2025). Summary Length Prediction using Image Processing. figshare. Journal contribution. https://doi.org/10.6084/m9.figshare.28239365.v1.

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
