# Peer review of "Enhanced transformer for length-controlled abstractive summarization based on summary output area"

_PeerJ Computer Science, doi:10.7717/peerj-cs.2667_

## Round 0.1 · original submission · Major Revisions

The paper needs to go through substantial revisions to incorporate the recommendations and address the concerns of the reviewers. They are all consistent highlighting a dearth of justifications throughout, lack of depth in the analysis and the need to improve the writing, clarity and formatting.

Given the volume of revisions required, the assessment of this work is borderline between a rejection and major revisions. As we are recommending major revisions here, we do require substantial revisions before it can be considered for review again.

Reviewer 1 ·

Basic reporting

*Although the main aim of the paper could be of interest for the journal readers, it requires a major review on the writing and formatting since I noticed a lot of typing and grammatical mistakes (e.g. extra spacing or missing spacing and some spelling mistakes .

* Text in figure 1 is not clear , please redraw it.

Experimental design

* Author require more explanation of the output of the proposed methodology. needs more experiments to be done

* The implementation details should be clearly specified in terms of hardware and software used in the experiments.

* No comparison have been found in that manuscript with other relevant methodologies, that should be added for the authenticity of the proposed methodology.

Validity of the findings

Summarizing, although the overall concept is fine, the paper is very unprofessionally written and thus the reported results are very unconvincing. Therefore, I advise the Authors to rewrite and resubmit it once again.

Additional comments

I suggest a major revision with a more professional editing and explanation.

Reviewer 2 ·

Basic reporting

All comments have been added in detail to the last section.

Experimental design

All comments have been added in detail to the last section.

Validity of the findings

All comments have been added in detail to the last section.

Additional comments

Review Report for PeerJ Computer Science
(Enhanced transformer for length-controlled abstractive summarization based on summary output area)

1. Within the scope of the study, transformers were used for length-controllable abstractive summarization, in which the image processing stage was integrated with CNN/Daily Mail datasets.

2. In the introduction and related works section, the importance of the subject and abstractive text summarization studies are mentioned and the limitations and techniques in the literature are adequately mentioned along with table-1.

3. It is observed that many different studies have been conducted in the literature with the dataset used. The main differences of this study from the studies in the literature and its main contributions to the literature need to be expressed more clearly and in detail.

4. Details about the dataset amount and preprocessing processes should be added, and the reason for choosing this dataset should be explained in detail when compared to other datasets that can be used for similar problems in the literature. It should be clarified whether the dataset distribution (training, validation, testing) was chosen randomly or whether any changes were made to the original dataset.

5. The explanation and details of the flowchart in figure-2 adequately explain the basic image processing stages.

6. The results obtained in the study should be compared and interpreted in detail with similar studies in the literature.

7. The text-to-text transfer transformer in figure-3 and the generative pre-trained transformer approaches in figure-5, which are transformer-based length controlled architectures, are clearly expressed.

8. When looking at the literature, there are many different transformer-based deep learning models that can be used to solve the problem within the scope of this study, but it should be explained why T5 and GPT models are preferred. It is also recommended to use at least one more up-to-date state-of-the-art transformer model in terms of working depth.

As a result, although the study is important in terms of subject and application approach, it is recommended to pay attention to the above-mentioned parts.

Reviewer 3 ·

Basic reporting

The focus of this paper is to generate length-controlled abstractive summaries. The rationale for integrating image processing with natural language processing models is well-established.
The paper is written fairly. Overall, the study presents a promising approach to length-controlled summarization, there is room for improvement. The current MS lacks in many aspects
* Justification is missing
* Lacking concrete analysis and discussion of results and models
This research requires a lot more rigor as mentioned below

Experimental design

The description of image processing techniques lacks depth, particularly regarding the selection and implementation of specific OpenCV functions.
*The authors developed an algorithm for detecting the optimal area for limiting the summary. However, there is a need to discuss the accuracy and shortcomings.
* The process for computing optimal summary length could be elucidated further, with explicit formulas or algorithms to demonstrate the calculation method
* Authors have mentioned in MS about using different font and line spacing depending upon the detected area. Some details are missing, such as how the model would change the font size or line spacing, if a smaller area is detected.
* The fine-tuning process for both T5 and GPT models could be described in greater detail, including hyperparameter tuning strategies and validation procedures
* Equation 1 and Equation 2, both do not provide sufficient information. The authors should write the equations providing significant information.

Validity of the findings

* The scope of this research seems to be limited. There are already many length-controlled abstractive summarization approaches that exist and limit the length of summaries just by providing the count of tokens.
*Authors should perform the comparison of their work with some latest research work.
* A comparative analysis of computational efficiency and resource requirements between the T5 and GPT would provide valuable insights into their practical feasibility

Additional comments

Please carefully review and update the technical language of the manuscript.

---

## Round 0.2 · Major Revisions

Please address the major revisions raised by reviewer 3. Please make sure to upload a tracked changes document which will facilitate the review in the next round.

Reviewer 2 ·

Basic reporting

All comments have been added in detail to the last section.

Experimental design

All comments have been added in detail to the last section.

Validity of the findings

All comments have been added in detail to the last section.

Additional comments

Review Report for PeerJ Computer Science
(Enhanced transformer for length-controlled abstractive summarization based on summary output area)

Thank you for the revision. The responses to the comments and the changes in the paper have been examined in detail. When both the changes made and the final version of the paper are examined, it has a certain originality and the potential to contribute to the literature. For this reason, I recommend that the paper be accepted. I wish the authors success in their future projects and papers. Best regards.

Reviewer 3 ·

Basic reporting

please see the additional comments section

Experimental design

please see the additional comments section

Validity of the findings

please see the additional comments section

Additional comments

It seems that the authors have wrongly uploaded the track changes version, because i cannot see any changes highlighted in that version expect the acknowledgement statement. and it is very hard to track if the previous comments of mine has been addressed or not. therefore, it is recommend to lookinto it and upload the proper tracked document.


“The focus of this paper is to generate length-controlled abstractive summaries. The rationale for integrating image processing with natural language processing models is well-established.
The paper is written fairly. Overall, the study presents a promising approach to length-controlled summarization, there is room for improvement. The current MS lacks in many aspects
* Justification is missing
* Lacking concrete analysis and discussion of results and models
This research requires a lot more rigor as mentioned below”

“The description of image processing techniques lacks depth, particularly regarding the selection and implementation of specific OpenCV functions.”
“*The authors developed an algorithm for detecting the optimal area for limiting the summary. However, there is a need to discuss the accuracy and shortcomings.”

“* The process for computing optimal summary length could be elucidated further, with explicit formulas or algorithms to demonstrate the calculation method”

“* Authors have mentioned in MS about using different font and line spacing depending upon the detected area. Some details are missing, such as how the model would change the font size or line spacing, if a smaller area is detected.”
Comment:
“* The fine-tuning process for both T5 and GPT models could be described in greater detail, including hyperparameter tuning strategies and validation procedures”
“* Equation 1 and Equation 2, both do not provide sufficient information. The authors should write the equations providing significant information.”

Comment:
“* The scope of this research seems to be limited. There are already many length-controlled abstractive summarization approaches that exist and limit the length of summaries just by providing the count of tokens.”
“*Authors should perform the comparison of their work with some latest research work.”
“* A comparative analysis of computational efficiency and resource requirements between the T5 and GPT would provide valuable insights into their practical feasibility”
Please carefully review and update the technical language of the manuscript.”

---

## Round 0.3 · accepted · Accept

Thank you for addressing reviewer concerns. Based on the reassessment of two reviewers, I can now recommend acceptance of this submission in its current form.

Reviewer 1 ·

Basic reporting

No comment

Experimental design

No comment

Validity of the findings

No comment

Additional comments

ow this paper have incorporated the solution of all necessary requirements and they have worked on modifying the paper in the light of the recommendations. Format along with style of the paper is fine now.

Reviewer 3 ·

Basic reporting

The author has addressed my previous concerns, therefore, I have no more comments.

Experimental design

The author has addressed my previous concerns, therefore, I have no more comments.

Validity of the findings

The author has addressed my previous concerns, therefore, I have no more comments.

Additional comments

The author has addressed my previous concerns, therefore, I have no more comments.